# Patient perspectives on physician competence: Validation of the CanMEDS framework

Ahmad A. El Lakis[1‡*], Ahmad El Issawi[2‡], Jana Al Tahan[3], Pascale Salameh[4,5,6,7]

1 Department of Internal Medicine, American University of Beirut Medical Center, Beirut, Lebanon,
2 Department of Plastic and Reconstructive Surgery, Lebanese University Faculty of Medicine, Beirut, Lebanon, 3 Faculty of Medicine, Lebanese University, Beirut, Lebanon, 4 Faculty of Pharmacy, Lebanese University, Hadat, Lebanon, 5 Gilbert and Rose-Marie Chagoury School of Medicine, Lebanese American University, Beirut, Lebanon, 6 Department of Primary Care and Population Health, University of Nicosia Medical School, Nicosia, Cyprus, 7 Institut National de Santé Publique d'Épidémiologie Clinique et de Toxicologie-Liban (INSPECT-LB), Beirut, Lebanon

‡ These authors are co-first authors on this work.
* ahmadellakis2000@gmail.com

## Abstract

Competency frameworks such as CanMEDS anchor medical training and accountability, yet most validations privilege educators and seldom test whether patients recognize and prioritize the same architecture. We conducted a cross-sectional online survey (February–June 2025) of Lebanese adults (N = 403) to validate, from the patient perspective, the seven CanMEDS roles and to examine sociodemographic moderators of endorsement and rank priorities. A 103-item instrument combined five-point endorsements and forced ranking. Psychometric evaluation used polychoric exploratory factor analysis and confirmatory factor analysis with diagonally weighted least squares; reliability was estimated with Cronbach's α and McDonald's ω. Robust linear regressions (HC4) modeled domain scores, and multinomial logistic regression analyzed rank priorities. Exploratory analysis supported seven factors explaining 68.6% of variance. Confirmatory analysis showed excellent fit with strong loadings and high internal consistency (α ≥ 0.90; ω ≥ 0.91). Men endorsed the Medical Expert role less than women (β=−2.96, p = 0.032). Higher family income showed graded positive associations with Medical Expert (e.g., > \$3,000/month: +8.66 points, p = 0.008). Lower educational attainment predicted lower priorities for Professionalism, Leadership, and Scholarship. Rural respondents prioritized Medical Expert, Communication, and Leadership more than urban peers, whereas physician age and gender were not significant predictors. Embedding patient-derived signals into competency-based medical education—through curricular emphasis, assessment weights, and multisource feedback—may strengthen social accountability and alignment with community expectations. Future work should test longitudinal stability, cross-cultural measurement invariance, and higher-order or bifactor models to parse shared variance among closely related roles.

**Data availability statement:** Data is provided within the manuscript or supplementary information files.

**Funding:** The authors received no specific funding for this work.

**Competing interests:** The authors have declared that no competing interests exist.

## Introduction

Competency-based medical education (CBME) emerged in part to address persistent gaps between educational processes and patient outcomes by explicitly orienting training toward the development of observable, outcome-linked abilities that translate to meaningful improvements in care delivery [1,2]. Its foundational premise is that by deconstructing professional performance into theoretically grounded yet contextually measurable domains, it becomes possible to design curricula, assessments, and feedback mechanisms that produce physicians who can meet evolving healthcare needs, including equity-sensitive, relational, and evidence-informed expectations [3–5]. Despite this progress, critical scholarship has cautioned against unexamined assumptions underlying CBME—specifically, that competencies are discrete, stable, and uniformly interpretable across contexts and raters—highlighting the need for continual validation of both the constructs and their measurement, particularly when extending frameworks into new stakeholder domains like patients [6–8].

Within this evolving landscape, CanMEDS has become a canonical and globally influential competency framework that explicitly codifies the integration of technical, relational, systemic, and ethical dimensions of physician practice. Originating in Canada and progressively adopted or adapted in numerous countries, CanMEDS articulates seven interdependent roles—Medical Expert, Communicator, Collaborator, Leader, Health Advocate, Scholar, and Professional—that collectively define a competent physician and serve both as curricular targets and assessment anchors [9]. The framework's theoretical strength lies in its rejection of reductionist proficiency models in favor of a multidimensional identity of medical practice, where expertise is inseparable from communication, collaboration, leadership, advocacy, scholarly engagement, and professional integrity—a perspective consonant with sociotechnical and patient-centered accounts of high-quality healthcare [10,11].

CanMEDS has been the subject of extensive implementation and validation efforts, particularly in postgraduate medical education, where it has informed workplace-based assessment systems, competency-by-design initiatives, and the development of national transformation models to ensure that trainees progress toward independence in a time-variable yet standards-consistent manner [2,3]. Foundational work has aimed to test whether the key competencies can serve as consistent outcome measures across different training stages and settings, interrogating their "fitness for purpose" and highlighting both the utility and the contextual dependencies of role observability and assessment [11,12]. The original development of competence inventories through Delphi processes and expert consensus further established content validity for workplace assessments, yet these validations have mostly centered on insider agreement and educator-driven definitions of "good" performance rather than on how patients perceive or prioritize these roles in practice [11].

Despite substantial uptake, the literature also recognizes complexities in competency modeling that bear directly on how frameworks like CanMEDS are interpreted, measured, and operationalized. Competencies do not exist in isolation; empirical work has repeatedly revealed overlapping constructs, with behaviors simultaneously

expressing multiple roles, which challenges simplistic factor structures and demands sophisticated psychometric techniques to disentangle shared and unique variance [10].

The existing validation ecosystem thus presents both a template and a gap: while CanMEDS has been refined for reliability and relevance within educational systems, the external alignment of its domains with patient values—particularly how patients differentiate, prioritize, and interpret overlapping physician roles—remains under-explored [7,13,14]. Patients are not passive recipients of competency deployment; they actively infer physician role performance through their interactions, expectations, and sociocultural lenses, making them potentially discerning evaluators capable of both recognizing common structures and revealing systematic blind spots when professional assumptions diverge from lived experience [14,15].

The omission of the patient perspective in much of the existing competency validation literature has left a critical blind spot, because what educators and trainers value in physician performance does not always align with what patients experience, prioritize, or interpret as quality care [7,10,14,16]. Studies of stakeholder discordance have shown that experts may emphasize conceptual or systemic aspects of competence while lay receivers of care attend more to relational, communicative, and pragmatic elements, producing divergence in perceived "good" medical practice unless the framework explicitly incorporates multiple vantage points [7,11,13,17]. Although the CanMEDS framework has been extensively studied from the perspective of physicians, educators, and accreditation bodies, the voice of patients remains comparatively underrepresented. Most competency frameworks are developed through expert consensus, yet patients—who are the ultimate recipients of care—rarely contribute to defining or weighting these competencies. This gap limits our understanding of how patients prioritize physician competencies and how these expectations may differ across cultural contexts.

Lebanon's university-based medical education system has progressively aligned with competency-based medical education (CBME) over the past decade. Notably, the American University of Beirut Medical Center (AUBMC) obtained ACGME-International accreditation, reflecting a broader regional shift toward outcomes-based training and external quality assurance [18,19]. Concurrent curriculum reforms within Lebanese programs have emphasized professional identity formation and the learning environment; at AUB, a seven-year longitudinal evaluation showed that targeted reforms were associated with higher learning-environment scores and improved empathy, key domains that map to widely used competency frameworks [20]. These efforts have unfolded amid overlapping national crises—economic collapse, the COVID-19 pandemic, and the Beirut port explosion—which strained clinical services, accelerated migration of health professionals, and forced rapid educational adaptations [21,22].

Prior Lebanese work on competencies has spanned both public expectations and learner assessment. At the undergraduate level, investigators at the Lebanese University validated a concise competency acquisition scale based on Englander's common taxonomy, demonstrating strong construct validity and internal consistency and offering a tool for tracking CBME implementation nationally [23]. Complementing these lines of inquiry, longitudinal curricular reform at AUB documented gains in empathy and other professionalism-adjacent outcomes [20], while studies of postgraduate trainees highlighted heavy wellbeing pressures that can erode competency development unless addressed by program-level supports [24]. Collectively, this literature shows that CBME in Lebanon is active, measurable, and increasingly tailored to societal and system needs.

The influence of patient sociodemographic and contextual characteristics on competency valuation is well established and provides a necessary layer to interpret patient-derived data; gender, socioeconomic status, educational attainment, rural versus urban domicile, and work background systematically shape what patients expect, how they interpret physician behavior, and which roles they prioritize in their own calculus of quality care [25,26].

The objective of this study was to develop and conduct a preliminary psychometric evaluation of a patient-adapted instrument assessing public perceptions of the seven CanMEDS roles. Specifically, we aimed to (i) examine the dimensionality of the instrument using exploratory and confirmatory factor analysis, (ii) assess internal consistency using reliability indices appropriate for ordinal data, and (iii) evaluate demographic differences in perceived role importance.

Based on theoretical grounding and prior empirical work, we hypothesized that patients would recognize a seven-factor structure corresponding to the CanMEDS roles. We further anticipate systematic demographic influences, such that higher income would correlate with greater weighting of the Medical Expert role, gender would modulate prioritization patterns, and rural patients would emphasize roles combining technical credibility with communication and leadership, while observable physician attributes like age and gender would exert smaller effects on the underlying hierarchy of patient-valued competencies. Finally, we posited that combining endorsement and ranking information would yield a richer, more discriminating portrait of patient values than either approach alone, allowing for detection of nuanced trade-offs and reinforcing contextualized interpretation of competency importance.

## Methods

### Ethical information

The study protocol was reviewed and approved by the Ethics Committee of the Lebanese Hospital Geitaoui- University Medical Center on February 5th, 2025. Data was collected between the months of February and June 2025.

All procedures adhered to the ethical standards of the 1964 Helsinki Declaration and its later amendments. Before accessing the questionnaire, every respondent read an electronic information sheet describing the study purpose, data handling, confidentiality safeguards, and voluntariness. Only individuals who ticked the mandatory consent box were allowed to fill the survey; no identifiers were recorded, guaranteeing anonymity. Respondents were advised that they can terminate the survey at any point without explanation. No monetary incentives were offered.

### Study design

We conducted a cross-sectional observational study. Data were collected exclusively online to maximise geographic reach. The study followed STROBE recommendations for reporting observational research.

### Study population

The study population included Lebanese patients who had visited at least one physician during their lifetime. Inclusion criteria included being of Lebanese residency, age above 18 years, absence of known cognitive impairment, and at least one lifetime visit to any physician. Exclusion criteria included refusing to fill the survey. Based on a power analysis in G*Power 3.1 (small expected effect, $f^2 = 0.05$; $\alpha = 0.05$; power $= 0.80$; 20 predictors), a minimum sample size of 415 participants was required.

### Data collection

The survey was filled electronically using Google Forms. The survey link was disseminated through social-media channels (primarily WhatsApp) using the snowball sampling technique, where participants are encouraged to forward the survey link to their family and friends who in turn forward it to their family and friends. To contextualize representativeness, we compared sample demographics with national statistics from the Central Administration of Statistics. The sample included a higher proportion of university-educated respondents than the general population, consistent with known digital access patterns in Lebanon. As such, findings should be interpreted as preliminary psychometric evidence rather than population-level inference. Raw responses were downloaded to an encrypted, password-protected file accessible only to the analytic team.

### Measures

Item generation drew on three sources: validated sub-scales (the 20-item TCom-skill GP scale for Communication [9,27] and the 43-item Penn State Professionalism Questionnaire [28]), published Lebanese CanMEDS survey [29] for Medical Expert, Collaborator, Leader, Scholar and Health Advocate items, and the general literature.

The survey included general demographic and socioeconomic questions followed by several questionnaires pertaining to the CanMEDS domains. The core of the instrument comprised endorsement and ranking tasks for the seven CanMEDS domains. For the endorsement section, participants rated on five-point Likert scales (1 = Strongly disagree to 5 = Strongly agree) the extent to which they believed physicians should exhibit each role's attributes. The Medical Expert domain included seven items (e.g.,; "Preserves the patient's safety"), while Communicator consisted of twenty items (e.g.,; "Lets me ask questions"). The Health Advocate domain featured four items (e.g.,; "Actively addresses barriers you face in accessing care"), Collaborator had five items (e.g.,; "Involves you and your family in decision-making"), and Scholar comprised six items (e.g.,; "Discusses new research findings relevant to your condition"). The extensive Professional domain included forty-three items (e.g.,; "Reports data consistently, accurately and honestly"), and Leader was measured with five items (for example, item 5: "Efficiently manages time during consultations"). Domain scores were computed as the mean of items. Higher score translated to a higher endorsement of our description of each domain.

Item generation followed a multi-stage process. First, the authors mapped each of the seven CanMEDS roles to a pool of behaviorally anchored statements adapted from the official CanMEDS framework. A panel of five experts in medical education and public health evaluated content validity (CVI) for clarity, representativeness, and cultural appropriateness. During EFA, items were retained if they met all three criteria: [1] loading ≥ 0.40 on the primary factor, (2) cross-loading ≤ 0.30 on other factors, and [3] conceptual alignment with the intended CanMEDS domain. CFA was conducted on the final 7-factor solution using DWLS estimation, with model fit evaluated using CFI, TLI, RMSEA, and SRMR.

Following the endorsement section, participants completed a forced-rank task in which they rated the seven domain labels into order of importance (1 = most important to 7 = least important).

## Statistical analysis

Categorical variables were summarized using frequencies and percentages, while continuous variables were reported as means and standard deviations. Comparisons between male and female participants were performed using Pearson's Chi-squared test for categorical variables and Welch's two-sample t-test for continuous variables.

For psychometric analysis, dimensionality was assessed using exploratory factor analysis (EFA) on the polychoric correlation matrix of the 103 Likert-scale items. The number of factors to extract was determined based on parallel analysis and visual inspection of scree plots. EFA was performed using principal axis factoring with oblique (Promax) rotation. We retained items on their primary factor if the loading was ≥ 0.40 and considered secondary loadings negligible if <0.30. We reported factor loadings, eigenvalues, sum-of-squared (SS) loadings, proportion variance, and cumulative variance explained by the factor structure.

To validate the proposed measurement structure, we conducted confirmatory factor analysis (CFA), specifying a seven-factor model corresponding to the CanMEDS domains. The CFA allowed us to resolve these ambiguities and confirm a structurally coherent seven-role solution that aligned with the intended CanMEDS framework. Given the ordinal nature of the data, we used diagonally weighted least squares estimation (DWLS). We evaluated model fit using conventional criteria: Comparative Fit Index (CFI) and Tucker–Lewis Index (TLI) values ≥0.95 indicating excellent fit, root mean square error of approximation (RMSEA) ≤0.06 with associated 90% confidence intervals, and standardized root mean square residual (SRMR) ≤0.08. We reported $\chi^2$ statistics, degrees of freedom, p-values, standardized factor loadings, and inter-factor correlations.

Internal consistency was assessed using CFA-based reliability estimates appropriate for ordinal data. For each CanMEDS domain, a one-factor confirmatory factor model was estimated using the WLSMV estimator with all domain items specified as ordered categorical indicators. McDonald's omega ($\omega$), ordinal Cronbach's alpha ($\alpha_{ord}$), composite reliability, and average variance extracted (AVE) were computed using the semTools::reliability() function. Following psychometric recommendations, omega total was used as the primary reliability index, as hierarchical omega may be inflated in large-item unidimensional models.

To examine demographic and physician-related predictors of participants' prioritization of physician roles, we conducted multinomial logistic regression analyses for each of the seven role ranking outcomes. Endorsement outcomes were the seven domain scores scaled to 0–100. We fit separate multivariable robust linear regressions per domain using HC4 heteroskedasticity-consistent standard errors. Predictors were: participant gender (male/female), age (continuous), education (Bachelor, Secondary/lower, Postgraduate), work type (business, engineering, health, other), region (urban/rural), and monthly family income (<$250, $500–$1000, $1000–$3000, >$3000). Doctor-level covariates captured the respondent's usual doctor's gender and age group. We assessed collinearity via VIFs (<4 for all models), inspected residual-vs-fitted plots for heteroskedasticity (hence HC4), and confirmed linearity of continuous terms with component-plus-residual plots; prespecified two-sided α = 0.05 guided inference without post-hoc model selection.

To explore whether participant characteristics were associated with differences in role prioritization, we conducted simple descriptive comparisons of mean rank positions across key demographic subgroups. For each role, rank values ranged from 1 (highest priority) to 7 (lowest priority). Mean ranks were calculated separately for gender, region of residence, education level, hospital preference, and household income. These analyses were intended as exploratory and descriptive rather than inferential. No regression modeling or adjusted analyses were performed, as the primary aim was to assess whether any subgroup exhibited substantial deviations from the overall ranking pattern identified by the Plackett–Luce model.

All analysis was run on RStudio (R version 4.5.1). Statistical significance was considered when p-value was below a threshold of 0.05.

## Results

### Sample characteristics

A total of 403 participants completed the survey (S1 Table). The study included 403 participants, of whom 290 (72%) were female and 113 (28%) were male. Significant gender differences were observed in education level (p < 0.001), chronic visit status (p = 0.001), work type (p = 0.010), and preferred doctor gender (p < 0.001). For instance, males were more likely to hold an MD (26% vs. 9%) and preferred male doctors more often than females (27% vs. 14%). No significant differences were found in region, income, or preferences for doctor religion, language, or hospital type. Satisfaction ratings for physician roles (e.g., communicator, leader, scholar) were generally high in both groups, with no statistically significant differences across domains.

### Psychometric Properties

The exploratory factor analysis (EFA) (Table 1) revealed a seven-factor structure explaining approximately 68.6% of the total variance, broadly reflecting the intended CanMEDS domains: Medical Expert, Communicator, Health Advocate, Collaborator, Professionalism, Leader, and Scholar. While most items loaded strongly onto their expected factors, several concerns were noted. Cross-loadings emerged, particularly for items A6 and A7, which loaded onto both Medical Expert and Scholar, and E19 and E30, which loaded onto both Professionalism and Health Advocate. Additionally, the Leader and Scholar factors demonstrated relatively low explained variance and weaker correlations with other factors, raising concerns about their distinctiveness. Some items intended to measure leadership clustered closer to Communicator.

To address these limitations, a confirmatory factor analysis (CFA) (Table 2) was conducted, specifying the hypothesized seven-factor model. The CFA showed excellent fit to the data, with χ²(3,633) = 6,279.6, p < 0.001, CFI = 0.999, TLI = 0.999, RMSEA = 0.039 (90% CI: 0.038–0.041), p [RMSEA ≤ 0.05] = 1.000, and SRMR = 0.040. This confirmatory approach resolved the cross-loading issues, strengthened the representation of weaker domains, and demonstrated consistently high standardized factor loadings across all constructs.

Table 1. Exploratory factor analysis (EFA) results.

| Item | Profes-sionalism | Commu-nicator | Medical Expert | Collab-orator | Leader | Scholar | Health Advocate |
|---|---|---|---|---|---|---|---|
| **A. Medical Expert** | | | | | | | |
| Has solid knowledge and applies it to offer the best care | | | 0.804 | | | | |
| Is able to get a good medical history, perform a rigorous physical exam, asks for the necessary paraclinical tests, puts the right diagnosis and proposes to the patient a clear management of his illness based on priorities | | | 0.866 | | | | |
| Prescribes the appropriate treatment and explains about its side-effects | | | 0.882 | | | | |
| Ensures a continuity in the care and treatment of the patient | | | 0.883 | | | | |
| Preserves the patient's safety | | | 0.971 | | | | |
| Your doctor explains the evidence or reasoning behind their medical decisions | | | 0.761 | | | 0.406 | |
| Your doctor uses visual aids or diagrams to help explain your condition when needed | | | 0.779 | | | 0.562 | |
| **B. Communicator** | | | | | | | |
| Takes time to listen to me | | 0.552 | 0.321 | | | | |
| Does everything to make me feel I can trust him/her | | 0.737 | | | | | |
| Explains what the treatment is for | | 0.763 | | | | | |
| Takes account of my preferences in prescribing medication | | 0.809 | | | | | |
| Gives me the impression he/she has respect for me | | 0.841 | | | | | |
| Gives me information on the side effects of medication | | 0.813 | | | | | |
| Emphasises which are the most important drugs | | 0.808 | | | | | |
| Discusses any difficulties I have in complying with the treatment | | 0.813 | | | | | |
| Explains things in simple words | | 0.793 | | | | | |
| Offers new treatment | | 0.733 | | | | | |
| Writes the prescription legibly | | 0.895 | | | | | |
| Lets me ask questions | | 0.829 | | | | | |
| Gives me incentives to comply with the treatment | | 0.782 | | | | | |
| Gives me advice on prevention (diet, physical activity) | | 0.665 | | | | | |
| Gives the impression he/she knows his/her job | | 0.722 | | | | | |
| Communicates with the patient and the family with respect and compassion and leads a good conversation | | 0.742 | | | | | |
| Listens to the patient without interrupting and gives the necessary time to get the important information | | 0.828 | | | | | |
| Explains to the patient the disease and treatment | | 0.638 | | | | | |
| Encourages the patient and the family to ask questions to understand more the disease and take part in the decisions | | 0.734 | | | | | |
| Documents all information while preserving confidentiality | | 0.725 | | | | | |
| **C. Health Advocate** | | | | | | | |
| Works at the level of patients to ensure the prevention and awareness of diseases | | | | 0.422 | | | |
| Works at the level of the community to ensure the prevention and awareness of diseases | | | | 0.440 | | | |
| Your doctor discusses how to prevent your condition or maintain your overall health | | | | 0.438 | | | |
| Your doctor actively addresses barriers you face in accessing healthcare services (e.g., cost, transportation) | | | | 0.407 | | 0.333 | |

*(Continued)*

| Item | Profes-sionalism | Commu-nicator | Medical Expert | Collab-orator | Leader | Scholar | Health Advocate |
|---|---|---|---|---|---|---|---|
| **D. Collaborator** | | | | | | | |
| Collaborates effectively with other healthcare professionals | | | | 0.771 | | | |
| Knows how to deal with conflicts and misunderstandings with colleagues | | | | 0.759 | | | |
| Is capable of transferring the care of the patient to another colleague if necessary | | | | 0.861 | | | |
| Your doctor actively involves other healthcare professionals (e.g., nurses, specialists) to improve your care | | | | 0.919 | | | |
| Your doctor involves you and your family in decision-making about your treatment plan | | | | 0.571 | | | |
| **E. Professional** | | | | | | | |
| Upholds scientific standards and bases decisions on scientific evidence and experience | 0.592 | | | | | | |
| Maintains relationships with research subjects that do not exploit personal financial gain, privacy, or sexual advantages | 0.689 | | | | | | |
| Takes time to review other colleagues' work and provides meaningful and constructive comments to improve it | 0.811 | | | | | | |
| Seeks self improvement | 0.684 | | | | | | |
| Reports data consistently, accurately and honestly | 0.762 | | | | | | |
| Avoids offensive speech that offers unkind comments and unfair criticisms | 0.688 | | | | | | |
| Shows a willingness to initiate and offer assistance toward a colleague's professional and personal development | 0.873 | | | | | | |
| Promotes the welfare and development of junior faculty | 0.899 | | | | | | |
| Refusal to violate one's personal and professional code of conduct | 0.733 | | | | | | |
| Appreciates and respects the diverse nature of research subjects and/or patients, and honors these differences in one's work with them | 0.799 | | | | | | |
| Attends faculty meetings, seminars, and student research presentations as a reflection of support | 0.883 | | | | | | |
| Works collaboratively and respectfully within a team to the benefit of improved patient care or to the contribution of research | 0.866 | | | | | | |
| Participates in corrective action processes toward those who fail to meet professional standards of conduct | 0.898 | | | | | | |
| Does not seek to advance one's career at the expense of another's career | 0.692 | | | | | | |
| Volunteers one's skills and expertise for the welfare of the community | 0.811 | | | | | | |
| Meets commitments and obligations in a conscientious manner | 0.880 | | | | | | |
| Respects the rights, individuality, and diversity of thought of colleagues and students | 0.939 | | | | | | |
| Meaningfully contributes to the teaching mission of the department and the College of Medicine | 0.914 | | | | | | |
| Shows compassion | 0.548 | | | | | | 0.498 |
| Demonstrates adaptability in responding to changing needs and priorities | 0.876 | | | | | | |
| Promotes justice in the bio-medical science system by demonstrating efforts to eliminate discrimination in research | 0.998 | | | | | | |
| Respects patient autonomy and helps them make informed decisions | 0.809 | | | | | | |
| Assumes leadership in research endeavors | 0.988 | | | | | | |
| Recognizes one's own limitations | 0.802 | | | | | | |
| Assumes personal responsibility for decisions regarding research activities | 0.878 | | | | | | |
| Participates in activities aimed at attaining excellence in biomedical science | 1.050 | | | | | | |
| Reports medical or research errors | 0.936 | | | | | | |

*(Continued)*

| Item | Profes-sionalism | Commu-nicator | Medical Expert | Collab-orator | Leader | Scholar | Health Advocate |
|---|---|---|---|---|---|---|---|
| Acts in ways that show a commitment to confidentiality | 0.809 | | | | | | |
| Adopts uniform and equitable standards for research | 0.875 | | | | | | |
| Demonstrates empathy | 0.572 | | | | | | 0.511 |
| Advocates the patient's or research subject's interest over one's own interest | 0.612 | | | | | | 0.393 |
| Discloses any conflicts of interest in the course of professional duties and activities | 0.813 | | | | | | |
| Is professionally attired in a manner that is respectful of others | 0.735 | | | | | | |
| Responds to constructive criticism by working to improve one's capability in the area criticized | 0.857 | | | | | | |
| Commits to implement cost-effective research methods | 0.828 | | | | | | |
| Represents information and actions in a truthful way | 0.837 | | | | | | |
| Acts with his patients with high ethics | 0.767 | | | | | | |
| Acts in response to the society's expectation of professionalism | 0.748 | | | | | | |
| Follows the laws of the medical profession | 0.709 | | | | | | |
| Preserves his well-being in order to give the best care to patients | 0.726 | | | | | | |
| **F. Leader** | | | | | | | |
| Applies a policy of improvement in his care for patients | | 0.568 | | | | | |
| Ensures best quality with minimal use of resources | | 0.547 | | | | | |
| Leads well to ensure best quality of care | | 0.712 | | | | | |
| Manages well his time and work | | 0.821 | | | | | |
| Your doctor efficiently manages their time during your consultations | | 0.663 | | | | | |
| **G. Scholar** | | | | | | | |
| Reads regularly and follows a plan for continuous education | | 0.882 | | | | | |
| Teaches students without jeopardizing the patient's safety | | 0.779 | | | | | |
| Is up to date in his medical knowledge | | 0.824 | | | | | |
| Is involved in research | | 0.814 | | | | | |
| Your doctor discusses new treatments or research findings relevant to your condition | | 0.781 | | | | | |
| Your doctor participates in clinical trials or research that could benefit patient care | | 0.756 | | | | | |

Standardized factor loadings were high across all domains, ranging from 0.741 to 0.964 (Table 3). Inter-factor correlations were moderate to high (r = 0.460–0.706), indicating that while the domains were interrelated, they represented distinct constructs.

Fig 1 presents the latent factor correlation matrix derived from the exploratory factor analysis (EFA). The heatmap visualizes the strength and direction of the correlations between the seven identified factors. Most factors show moderate-to-strong positive correlations (r = 0.58–0.74), indicating that higher scores in one domain are generally associated with higher scores in related competencies.

Internal consistency reliability was excellent across all domains (Table 4). Cronbach's α coefficients ranged from 0.899 for the Communicator domain to 0.987 for the Professional domain McDonald's omega total for the seven CanMEDS-derived domains ranged from **0.91 to 0.99**, indicating excellent reliability. For Domain E, recalculated omega total was **0.987**, with an ordinal α of **0.992** and an average variance extracted (AVE) of **0.774**, demonstrating strong internal consistency and convergent validity.

**Table 2. Confirmatory factor analysis (CFA) fit indices. CFI = confirmatory fit indices; TLI = tucker-lewis index; RMSEA = root mean square error of approximation; SRMR = standardized root-mean-square residual.**

| Statistic | Value |
|---|---|
| χ² (df) | 6,279.6 (3,633) |
| p-value | <0.001 |
| CFI | 0.999 |
| TLI | 0.999 |
| RMSEA (90% CI) | 0.039 (0.038–0.041) |
| p [RMSEA ≤0.05] | 1.000 |
| SRMR | 0.040 |

**Table 3. Standardized factor loadings by domain.**

| Domain | Min Loading | Max Loading |
|---|---|---|
| Medical Expert | 0.741 | 0.964 |
| Communicator | 0.752 | 0.942 |
| Health Advocate | 0.755 | 0.935 |
| Collaborator | 0.761 | 0.921 |
| Professional | 0.772 | 0.950 |
| Leader | 0.748 | 0.932 |
| Scholar | 0.752 | 0.944 |

## Inferential analysis

Across domains, several consistent patterns emerged. Male participants endorsed the importance of the Medical Expert role significantly less strongly than female participants (β = −2.96, p = 0.032), with similar but non-significant negative trends across other domains (Table 5). Family income was positively associated with stronger endorsement of Medical Expert competencies, with participants earning >3000 USD per month rating its importance 8.66 points higher (p = 0.008), and those in the 1000–3000 USD and 500–1000 USD brackets also showing significant positive associations (β = 7.03, p = 0.016; β = 6.73, p = 0.018, respectively).

In the Communicator domain, no demographic predictors reached statistical significance, though positive trends were observed for participants with higher education levels. In the Health Advocate and Collaborator domains, male gender was again associated with lower (though non-significant) importance ratings, while rural residence tended to predict higher Health Advocate ratings (β = 8.63, p = 0.22), though this association was not statistically significant.

To analyze the complete ranking data, we fitted a Plackett–Luce model, the gold-standard probabilistic framework for rank outcomes. The model estimates a set of latent "worth" parameters for each role, representing the probability that a given role is preferred over another across all implied pairwise comparisons. Higher worth indicates higher perceived importance.

Table 6 presents the estimated worth parameters and 95% confidence intervals for the seven physician roles. The Medical Expert role had the highest worth and was therefore used as the reference category (worth = 1.00). All other roles were less likely to be ranked above Medical Expert.

Across respondents, the roles ranked immediately after Medical Expert were Communicator (worth = 0.69, 95% CI 0.59–0.82) and Professional (0.70, 95% CI 0.58–0.82), reflecting strong emphasis on interpersonal and ethical competencies. These roles were followed closely by Leader/Manager (0.68, 95% CI 0.58–0.80). The remaining roles—Health Advocate (0.60, 95% CI 0.51–0.71), Collaborator (0.55, 95% CI 0.47–0.65), and Scholar (0.52, 95% CI 0.43–0.61)—had notably lower worth parameters, indicating they were less frequently placed at the top of participant rankings.

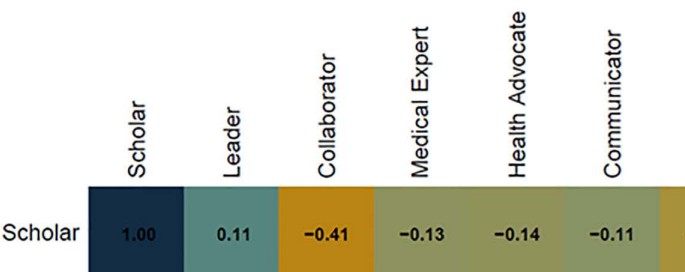

**Fig 1. Correlation matrix of the 7 factors of our study.**

Table 4. Internal consistency reliability.

| Domain | Cronbach's α | McDonald's ω |
|---|---|---|
| Medical Expert | 0.921 | 0.932 |
| Communicator | 0.899 | 0.910 |
| Health Advocate | 0.905 | 0.917 |
| Collaborator | 0.912 | 0.923 |
| Professional | 0.987 | 0.987 |
| Leader | 0.924 | 0.936 |
| Scholar | 0.915 | 0.926 |

To explore demographic variation in ranking patterns, we compared mean rank positions (1 = highest priority; 7 = lowest) across key respondent characteristics. Overall, subgroup differences were modest, and none altered the core hierarchy identified by the Plackett–Luce model.

**Gender.** Women and men showed highly similar preferences. Medical Expert was prioritized equally (female mean = 3.00; male = 2.95). Women placed slightly higher priority on Communicator (3.83 vs. 4.12) and Professional (3.98

**Table 5. Multinomial logit models results role ranks.**

| Predictor | Medical Expert (A) | Communicator (B) | Health Advocate (C) | Collaborator (D) | Professional (E) | Leader (F) | Scholar (G) |
|---|---|---|---|---|---|---|---|
| Male (vs Female) | **−2.96** (95% CI −5.7, −0.2), p=0.032 | −1.44 (95% CI −4.3, 1.5), p=0.33 | −2.96 (95% CI −6.9, 1.0), p=0.14 | −3.94 (95% CI −7.9, 0.1), p=0.05 | −1.57 (95% CI −4.9, 1.7), p=0.35 | −2.15 (95% CI −5.9, 1.6), p=0.25 | −2.75 (95% CI −6.1, 0.6), p=0.11 |
| Secondary/ Lower Educ (vs Bachelor) | — | — | — | — | **−24.6** (95% CI −30.0, −19.2), p<0.001 | **−18.9** (95% CI −32.4, −5.4), p=0.009 | **−23.7** (95% CI −31.2, −16.2), p<0.001 |
| Engineering Work (vs Business) | +4.3 (95% CI −3.4, 12.0), p=0.26 | −4.2 (95% CI −12.2, 3.9), p=0.30 | −3.7 (95% CI −15.1, 7.7), p=0.51 | +0.2 (95% CI −9.0, 9.4), p=0.97 | **−9.6** (95% CI −18.6, −0.6), p=0.037 | −3.5 (95% CI −14.1, 7.1), p=0.51 | −9.1 (95% CI −18.4, 0.2), p=0.055 |
| Income 500–1000$ (vs<\$250) | **+6.7** (95% CI 1.1, 12.3), p=0.018 | +2.8 (95% CI −3.0, 8.6), p=0.36 | −0.9 (95% CI −8.7, 6.9), p=0.83 | +1.0 (95% CI −6.6, 8.6), p=0.81 | +2.8 (95% CI −4.0, 9.6), p=0.42 | +2.2 (95% CI −5.6, 10.0), p=0.57 | +4.5 (95% CI −2.3, 11.3), p=0.21 |
| Income 1000–3000$ (vs<\$250) | **+7.0** (95% CI 1.3, 12.7), p=0.016 | +3.0 (95% CI −2.9, 8.9), p=0.34 | −2.7 (95% CI −10.8, 5.4), p=0.53 | +1.2 (95% CI −6.7, 9.1), p=0.78 | +5.0 (95% CI −1.9, 11.9), p=0.16 | +2.4 (95% CI −5.6, 10.4), p=0.55 | +7.0 (95% CI −0.1, 14.1), p=0.05 |
| Income >3000$ (vs<\$250) | **+8.7** (95% CI 2.4, 15.0), p=0.008 | +2.8 (95% CI −4.0, 9.6), p=0.42 | +1.4 (95% CI −7.4, 10.2), p=0.78 | +5.0 (95% CI −3.7, 13.7), p=0.30 | +5.9 (95% CI −1.7, 13.6), p=0.14 | +4.4 (95% CI −4.6, 13.4), p=0.32 | +7.3 (95% CI −0.5, 15.1), p=0.07 |

**Table 6. Plackett–luce estimated worth parameters for physician roles.** *(Worth = relative probability of being ranked above other roles; Medical Expert=reference=1.00).*

| Role | Worth | 95% CI (Lower) | 95% CI (Upper) |
|---|---|---|---|
| Medical Expert (A) | **1.00** | **1.00** | **1.00** |
| Communicator (B) | 0.692 | 0.587 | 0.814 |
| Professional (C) | 0.695 | 0.590 | 0.819 |
| Leader (D) | 0.680 | 0.577 | 0.802 |
| Health Advocate (E) | 0.600 | 0.508 | 0.709 |
| Collaborator (F) | 0.551 | 0.466 | 0.652 |
| Scholar (G) | 0.516 | 0.434 | 0.614 |

**Interpretation:** A = most prioritized; G = least prioritized.

vs. 4.21), while men gave marginally higher priority to Leader (4.06 vs. 4.18). Differences for Health Advocate, Collaborator, and Scholar were minimal (<0.2 rank points).

**Region.** Urban and rural respondents demonstrated nearly identical patterns, with Medical Expert and Communicator consistently at the top. Rural respondents ranked Medical Expert slightly higher (2.79 vs. 3.06), whereas urban respondents showed marginally higher preference for Communicator (4.01 vs. 3.66). Participants from mixed areas displayed more variability due to small sample size but did not deviate from the general trend.

**Education.** Rank patterns were broadly stable across educational levels. Participants with higher degrees (master's/PhD) tended to prioritize Medical Expert and Communicator slightly more, while those with bachelor's or undergraduate degrees ranked Scholar and Collaborator somewhat lower. These shifts remained small and did not change the overall ordering.

**Hospital Preference.** Respondents preferring private or public hospitals showed minor differences. Those preferring private care ranked Medical Expert (2.98) and Communicator (3.96) similarly to those with no preference (2.94 and 3.80).

The small subgroup preferring public hospitals showed greater variability (e.g., C_mean = 2.67), reflecting limited sample size rather than a systematic shift.

**Income.** Rank patterns were stable across income brackets. Lower-income respondents (<250 USD) showed slightly higher priority for Leader (4.12) and Scholar (4.79), while higher-income respondents (>3000 USD) placed somewhat greater emphasis on Leader (3.58). However, differences remained small, and Medical Expert and Communicator consistently remained the top-ranked roles across all income levels.

## Discussion

This study set out to validate the CanMEDS physician competency framework from the perspective of patients and community members in Lebanon. In competency-based medical education, most frameworks are defined by experts and institutions, yet patients are the ultimate stakeholders in how these roles are enacted in practice. By adapting the seven CanMEDS roles into a patient-facing instrument and evaluating its psychometric properties, we provide empirical evidence on how the public conceptualizes and prioritizes physician competencies. Our main findings can be summarized as follows. First, the instrument demonstrated a clear seven-factor structure that closely mirrored the canonical CanMEDS roles, with excellent model fit and high internal consistency across all domains. Second, patients consistently prioritized the Medical Expert and Communicator roles, while still recognizing the importance of Professional, Leader, Health Advocate, Collaborator and Scholar roles. Third, sociodemographic predictors such as gender, income, education, work type, and rural versus urban residence were associated with subtle but interpretable differences in how roles were valued. Together, these findings suggest that the CanMEDS framework is both recognizable and meaningful when viewed through the patient lens, while also revealing where expectations vary across patient subgroups.

The high fit indices of the seven-factor CFA confirm the construct validity of the patient-adapted instrument and align with the theoretical hypothesis that CanMEDS roles are a comprehensive yet distinct constellation of professional competencies [30]. Such model performance is rare when a complex framework is presented as-is to a lay audience, as previous attempts to map multifaceted health constructs onto non-expert populations often require trimming or rewording to reach acceptable fit [31]. Our ability to preserve all seven roles intact suggests that the underlying boundaries—between, for example, Leader and Collaborator or Scholar and Medical-Expert—are sufficiently intuitive for patients [32]. The process by which cross-loadings were handled is informative: during EFA, several Scholar items loaded secondarily on Medical-Expert, while some Health-Advocate items drifted toward Professionalism; this was unsurprising given evidence that patients frequently interpret evidence-based practice as integral to clinical expertise [33].

The pattern of gender differences in domain endorsement, most notably that males placed relatively less emphasis on the Medical Expert role, likely reflects a complex interplay of sociocultural expectations, differential health literacy, and gendered perceptions of clinical authority and interpersonal style [34–37]. Prior work has shown that gender norms shape how patients interpret and trust medical advice, with men sometimes expressing different thresholds and explanations for what they consider "expert" behavior and being less likely to engage in shared decision-making unless expertise is framed in certain culturally agreeable ways [15,34]. Furthermore, health literacy, which differs systematically by gender in some contexts, adjusts the perceived prominence of technical competence, such that lower active engagement or differential comprehension can reduce the weighting of the Medical Expert role unless it is communicated in accessible formats [37–39]. In addition, the literature on gendered communication and evaluation reveals that female physicians are more often described with "communal" attributes that correlate with higher patient ratings, suggesting that male patients may conceptualize expertise differently and that their internal benchmarks for "expert" performance may be less tied to overt technical displays than to other cues [34,40,41].These gendered perceptual frameworks likely contribute to the difference we observed in how the Medical Expert domain is prioritized across male and female respondents.

Socioeconomic status, particularly higher income, was associated with greater prioritization of the Medical Expert role, consistent with previous evidence that patients of higher socioeconomic status hold higher expectations for technical

competence and are more assertive in evaluating clinician knowledge and diagnostic certainty [42–44]. It is also important to consider that wealthier patients often have more consistent exposure to smoothly functioning healthcare systems, leading them to conceptualize quality care predominantly through the lens of demonstrable expertise, whereas lower-income individuals may have experienced barriers, perceived differential treatment, or bias, shifting their valuation toward domains such as advocacy or communication as safeguards against marginalization [39]. These dynamics are amplified by the fact that lower-SES patients may internalize healthcare systems' differential responsiveness and bias towards them and give priority to roles that feel more immediately protective, such as Health Advocate or Communicator, rather than the more abstract technical mastery [42,44,45]. In contrast, patients with higher economic capital are more likely to seek confirmation of competence and may interpret expertise as a signal of worthiness of monetary investment and trust [42,43].

Education and work-type showed strong influences on how certain CanMEDS domains were valued, with participants possessing lower formal education or coming from non-managerial/manual occupations placing less emphasis on Professionalism, Leadership, and Scholar roles. This observation aligns with the literature showing that education shapes health literacy, the ability to understand professional language, and the frameworks through which patients assess abstract competencies like scholarship or systemic leadership [38,39,46,47]. Individuals with higher education frequently engage more as informed partners in care, have better familiarity with evidence-based thinking, and thus are more aware of the value of continuous learning and leadership in clinical contexts [15,37,48].

The differences between work-type groups, such as engineering versus business backgrounds, likely reflect variations in professional socialization; those from environments emphasizing hierarchical coordination or strategic decision-making may have a more developed schema for recognizing and appreciating physician leadership, while others more involved in task-oriented or localized roles may not spontaneously recognize such abstract competencies without contextual framing [49]. These differences underscore the need to tailor physician communication so that the Scholar role, for instance, is made recognizable through explicit articulation of evidence behind recommendations, and leadership is contextualized in patient-relevant terms, thereby reducing barriers for patients with diverse educational and occupational backgrounds [14,16].

Unexpectedly, physician-specific attributes such as age and gender did not significantly predict patient ranking of the core competency domains which suggests that patients' own identity, experience, and interpretation dominate over superficial provider characteristics when determining the relative importance of physician roles [34,50]. While some literature emphasizes the importance of concordance (e.g., gender or cultural matching) for trust and satisfaction, other studies have found that deeper patient-level factors shaped by socioeconomic background, education, and previous healthcare interactions mediate how competency constructs are perceived [11,15]. This suggests that while physician demographics may influence specific interpersonal dynamics, they do not fundamentally re-order the hierarchy of competencies from the patient's perspective, supporting the idea that patient valuation is a result of internalized expectations rather than provider features [51].

Geographic context played a significant role in shaping role prioritization, with rural participants assigning greater importance to Medical Expert, Communicator, and Leader functions, consistent with the expectation that rural physicians often operate as indispensable generalists, leaders within fragmented care systems, and primary conduits of information for dispersed populations [52–54].

In rural settings, where specialty access is limited and patients frequently rely on a limited number of clinicians, the combination of deep technical competence, clear two-way communication to compensate for informational gaps, and leadership in care coordination becomes essential to meet diverse health needs, amplifying the noticeability of these domains relative to urban environments where role differentiation is more distributed across teams [53,55]. The importance of communication in these contexts is also deeply linked to trust formation, as rural patients with limited healthcare access rely heavily on interaction quality to interpret expertise, meaning that the Communicator role will scaffold acceptance of medical expertise and leadership decisions [52]. This recalibration of role hierarchy demonstrates the importance of local expectations in shaping what patients want from physicians and suggests that competency frameworks used in diverse settings must account for such subtle differences to avoid one-size-fits-all misalignment [11].

Our study has several limitations to be considered while interpreting the findings. the cross-sectional nature of the study limits causal inferences about what drives changes in role valuation or competency perception; correlations between patient characteristics and domain prioritization could reflect reverse causation or unmeasured confounding, a limitation intrinsic to snapshot observational designs that longitudinal approaches would better address. Without temporal sequencing, it remains uncertain whether, for example, educational exposure leads to elevated Scholar valuation or whether individuals predisposed to appreciating evidence-based reasoning seek educational experiences that reinforce that orientation. Self-report measurement introduces potential biases, including social desirability, where respondents may over-report what they perceive as valued by the system or under-report less socially approved preferences, thereby distorting the true hierarchy of internal valuations. Although steps can be taken to mitigate social desirability bias, such as assuring anonymity, residual distortion remains possible, particularly for constructs like Professionalism and Leadership that carry normative weight in public discourse. Generalizability may also be limited since our sample's geographic, cultural, or healthcare system context is not representative of broader populations. It is well known that regional healthcare norms, expectations about physician roles, and the cultural articulation of competencies can vary significantly across countries and systems, which might limit direct extrapolation without cross-cultural calibration. Systemic biases, including racial/ethnic disparities and medical mistrust, further complicates generalizability, as underrepresented groups may appraise and rank competencies differently due to lived experiences of discrimination or differential access. Lastly, the high inter-factor correlations observed—particularly clustering among Medical Expert, Health Advocate, and Professionalism—while theoretically coherent, raise the possibility of construct overlap or redundancy, implying that some dimensions may be manifestations of higher-order latent traits rather than fully distinct constructs.

## Conclusion

This study offers an initial psychometric evaluation of a patient-centered CanMEDS instrument in Lebanon. While further validation is required, including replication, invariance testing, and larger representative samples, the findings demonstrate the relevance and interpretability of the CanMEDS framework from the patient perspective. The findings underscore that patient values are demographically patterned, with gender, socioeconomic status, education, work-type, and geographic setting shaping how competencies are weighted, and these differences should inform adaptive curriculum design, assessment weighting, and equity-focused communication strategies. Looking forward, integrating patient-defined priority structures into competency development and evaluation offers a path toward more socially responsive medical training and practice, aligning professional standards with the nuanced expectations of diverse care recipients and thereby strengthening trust, relevance, and outcomes in healthcare delivery.

## Supporting information

**S1 Table. Participant baseline characteristics.**
(DOCX)

**S1 File. Questionnaire.**
(DOCX)

**S1 Data. Excel sheet containing the dataset used in this study.**
(XLSX)

## Author contributions

**Conceptualization:** Pascale Salameh.

**Writing – original draft:** Ahmad El Lakis, Ahmad El Issawi, Jana Al Tahan.

**Writing – review & editing:** Ahmad El Lakis, Ahmad El Issawi, Jana Al Tahan, Pascale Salameh.

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
