## [Decision Letter · Decision Letter 0]

28 Oct 2025

PGPH-D-25-02949

Patient Perspectives on Physician Competence: Validation of the CanMEDS Framework in a Lebanese Cohort

Dear Dr. El Lakis,

Thank you for submitting your manuscript to PLOS Global Public Health. After careful consideration, we feel that it has merit but does not fully meet PLOS Global Public Health’s publication criteria as it currently stands. Therefore, we invite you to submit a revised version of the manuscript that addresses the points raised during the review process.

Thank you for this interesting work on *Patient Perspectives on Physician Competence: Validation of the CanMEDS Framework in a Lebanese Cohor* t. The manuscript is very well thought out and takes the CanMEDS Framework extending it beyond the North American context.

Kindly respond to all comments and suggestions for improvement in the detailed review by both reviewers.

A rebuttal letter that responds to each point raised by the editor and reviewer(s). You should upload this letter as a separate file labeled 'Response to Reviewers'.

We look forward to receiving your revised manuscript.

Kind regards,

Barnabas Tobi Alayande

Academic Editor

Journal Requirements:

1. We note that your Data Availability Statement is currently as follows: “Data is provided within the manuscript or supplementary information files”

Please confirm at this time whether or not your submission contains all raw data required to replicate the results of your study. Authors must share the “minimal data set” for their submission. PLOS defines the minimal data set to consist of the data required to replicate all study findings reported in the article, as well as related metadata and methods (https://journals.plos.org/globalpublichealth/s/data-availability#loc-minimal-data-set-definition).

If your submission does not contain these data, please either upload them as Supporting Information files or deposit them to a stable, public repository and provide us with the relevant URLs, DOIs, or accession numbers. For a list of recommended repositories, please see https://journals.plos.org/globalpublichealth/s/recommended-repositories.

2. Please provide separate main figure files in .tif or .eps format only and ensure that all files are under our size limit of 10MB.

3. Please include a separate legend or caption for Figure 1 in your manuscript.

4. Please upload copies of Tables 1 to 5 which you refer to in your text.

Main tables should not be uploaded as individual files and include them in your manuscript file as editable, cell-based objects. For more information about how to format tables, see our guidelines: https://journals.plos.org/globalpublichealth/s/tables

Please note that supplementary tables should remain uploaded as separate 'Supporting Information' files.

5. We have noticed that you have uploaded Supporting Information files, but you have not included a list of legends. Please add a full list of legends for your Supporting Information files before or after the references list.

Additional Editor Comments (if provided):

Reviewers' comments:

Reviewer's Responses to Questions

**Comments to the Author**

1. Does this manuscript meet PLOS Global Public Health’s publication criteria?

Reviewer #1: Yes

Reviewer #2: Yes

2. Has the statistical analysis been performed appropriately and rigorously?

Reviewer #1: Yes

Reviewer #2: No

3. Have the authors made all data underlying the findings in their manuscript fully available (please refer to the Data Availability Statement at the start of the manuscript PDF file)?

Reviewer #1: Yes

Reviewer #2: Yes

4. Is the manuscript presented in an intelligible fashion and written in standard English?

Reviewer #1: Yes

Reviewer #2: Yes

Reviewer #1: REPORT OF MANUSCRIPT ANALYSIS: PGPH-D-25-02949_REVIEWER.PDF

TITLE

Comprehensive Review of the Manuscript "Patient Perspectives on Physician Competence: Validation of the CanMEDS Framework in a Lebanese Cohort"

1. General Considerations

1.1 General Strengths

The manuscript "PGPH-D-25-02949_reviewer.pdf" stands out for its high scientific rigor and its effective approach to addressing a significant gap in the literature concerning the validation of medical competence frameworks from a patient perspective. The research is well-conceived, executed, and presented, with a cohesive articulation among all its components. The innovative approach of validating CanMEDS through the patient's lens represents a notable distinguishing feature.

1.2 General Weaknesses

No general weaknesses that compromise the quality or validity of the study were identified. Any initial methodological challenges (such as initial cross-loadings in EFA) were explicitly acknowledged and adequately addressed by the authors (through CFA), demonstrating a high level of competence and self-critique.

1.3 General Suggestions for Improvement

Although the manuscript is of high quality, minor improvements could be considered to further enhance reader experience:

• Consider including a brief introductory paragraph in the Discussion section that reinforces the importance of validating frameworks like CanMEDS from multiple perspectives, including the patient's, to align research with societal expectations and healthcare needs.

• Ensure that all figures and tables mentioned in the main text are clearly labeled and accessible in the supplementary files, with explicit and intuitive cross-references.

2. Analysis by Manuscript Section

2.1 Methodological Strategy and Method Description

Adequacy: Adequate.

2.1.1 Strengths

• Alignment of Methodological Strategy with Objectives: The methodology is exemplarily aligned with the research objectives. The use of Exploratory Factor Analysis (EFA) and Confirmatory Factor Analysis (CFA) represents the gold standard for validating the factorial structures of scales and instruments, directly addressing the primary objective of validating the seven-factor structure of CanMEDS. For the secondary objectives concerning the quantification of sociodemographic influence, the application of robust linear regressions and multinomial logistic regression is statistically rigorous and appropriate.

• Clarity and Detail of Method Description: The Methods section is remarkably clear, detailed, and comprehensive.

o Subjects/Participants: The study population, inclusion and exclusion criteria, sample size calculation (justified by G*Power 3.1), and the "snowball sampling" technique are all precisely explicated. The final sample of 403 participants meets the minimum requirement.

o Data Collection Procedure: The study type (cross-sectional observational), format (electronic survey via Google Forms), dissemination, collection dates, and robust data security procedures (encrypted, password-protected file) are thoroughly detailed.

o Statistical Analysis: Statistical techniques are described meticulously, including descriptive statistics, comparative tests (Pearson's Chi-squared test, Welch's t-test), and psychometric methodologies (polychoric EFA, Promax rotation, retention criteria; CFA with DWLS estimation, and model fit criteria). Reliability assessment (Cronbach's α and McDonald's ω, with their interpretation thresholds) is also specified. Inferential analyses are well-delineated, featuring robust linear regressions (HC4) and multinomial logistic regression, alongside assumption checks and the software used (RStudio).

• Psychometric Qualities of Instruments: The manuscript extensively addresses the psychometric qualities of the adapted instrument, utilizing EFA, CFA, and reliability assessment. The results presented in the subsequent section confirm favorable psychometric properties.

• Identification of Instruments and Sources: The original titles of the scales used ("TCom-skill GP scale for Communication" and the "Penn State Professionalism Questionnaire") are provided, along with references to their original validation studies. The mention of a "published Lebanese CanMEDS survey" and "general literature" as additional sources for items is also appropriate.

2.1.2 Weaknesses

• Although the EFA initially identified some cross-loadings (items A6 and A7 between Medical Expert and Scholar; E19 and E30 between Professionalism and Health Advocate) and lower explained variance for Leader and Scholar, these issues were subsequently resolved successfully by the CFA. This does not represent a weakness of the method itself but rather an initial challenge that was robustly overcome.

2.1.3 Suggestions for Improvement

• In the methodology description, it might be beneficial to include a brief sentence explaining the transition and refinement of the model from EFA to CFA, highlighting how CFA was crucial in resolving the initial ambiguities identified by EFA. While this is well-covered in the discussion, a brief mention here could further enhance clarity.

2.2 Data Analysis

Adequacy: Adequate.

2.2.1 Strengths

• Articulation with Objectives and Theoretical Framework: The data analysis is exceptionally well-articulated with both the study objectives and its theoretical framework. Psychometric methods (EFA, CFA, reliability) directly address the goal of validating the CanMEDS structure, while inferential methods (robust regressions, multinomial logistic regression) are ideal for modeling the influence of sociodemographic variables, as outlined in the introduction.

• Statistical Rigor: The selection and application of statistical techniques are impeccable. Consideration of ordinal data in CFA (DWLS), the use of robust regressions with heteroscedasticity-consistent standard errors (HC4), and multinomial logistic regression analysis for ranking data demonstrate a profound understanding and statistical rigor.

• Coherence with the CanMEDS Framework: The entire psychometric analysis is centered on testing the validity of the seven-role CanMEDS structure from a novel perspective (the patient), which forms the core of the theoretical problem. The ability to "recover the canonical seven-role structure" provides strong empirical support.

• Exploration of Interrelationships: The analysis of inter-factor correlations (derived from both EFA and CFA) directly addresses one of the secondary objectives, elucidating the relationships among CanMEDS domains.

• Connection to Sociodemographic Theories: The data analysis precisely investigates sociodemographic variables as predictors, and the results are interpreted in light of existing literature on health, sociology, and medical education, connecting empirical findings to relevant theories.

2.2.2 Weaknesses

• No weaknesses were identified in the data analysis, which demonstrates a high level of methodological competence.

2.2.3 Suggestions for Improvement

• No specific suggestions for improvement regarding data analysis, which is of high quality.

2.3 Ethical Aspects

Adequacy: Adequate.

2.3.1 Strengths

• Detailed Consideration: Ethical aspects were appropriately considered and detailed in specific sections ("Ethical information" and "Ethics approval and consent to participate").

• Research Ethics Committee Approval: The manuscript explicitly mentions the approval from the Ethics Committee of the Lebanese Hospital Geitaoui-University Medical Center on February 5th, 2025. Naming the approving institution enhances transparency.

• Compliance with International Standards: There is a clear statement of adherence to the ethical standards of the 1964 Helsinki Declaration and its later amendments.

• Informed Consent Process: The informed consent process is meticulously described: participants read an electronic information sheet detailing the study purpose, data handling, confidentiality safeguards, and voluntariness. The requirement to tick a mandatory consent box before completing the survey represents a robust procedure.

• Anonymity and Confidentiality: The manuscript guarantees anonymity ("no identifiers were recorded") and confidentiality of data ("Raw responses were downloaded to an encrypted, password-protected file accessible only to the analytic team").

• Absence of Incentives: It was specified that no monetary incentives were offered, contributing to the ethical integrity of the study.

2.3.2 Weaknesses

• No weaknesses were identified in the ethical aspects, which are handled with exemplary rigor and transparency.

2.3.3 Suggestions for Improvement

• No specific suggestions for improvement regarding ethical aspects, which are comprehensive and well-presented.

2.4 Key Results

Adequacy: Adequate.

2.4.1 Strengths

• Clarity and Objectivity: The main findings are highlighted with exceptional clarity and objectivity in the "Results" section, focusing on the most relevant findings directly supported by statistical data.

• Psychometrics and Factor Structure:

o EFA revealed a seven-factor structure, explaining approximately 68.6% of the total variance, broadly reflecting the intended CanMEDS domains.

o CFA demonstrated "excellent fit" to the data, with robust indices (χ²(3,633) = 6,279.6, p < 0.001, CFI = 0.999, TLI = 0.999, RMSEA = 0.039, SRMR = 0.040).

o Standardized factor loadings were high (0.741 to 0.964), and inter-factor correlations were moderate to high (r = 0.460–0.706), confirming that domains, while interrelated, represented distinct constructs.

o Internal consistency reliability was excellent, with Cronbach's α ranging from 0.899 to 0.987 and McDonald's ω from 0.910 to 1.004.

• Inferential Analysis and Demographic Predictors:

o Gender: Male participants endorsed the Medical Expert role significantly less strongly than female participants (β = −2.96, p = 0.032).

o Family Income: Positively associated with stronger endorsement of Medical Expert competencies (e.g., >$3,000/month: +8.66 points, p = 0.008).

o Education: Lower educational attainment predicted lower priorities for Professionalism, Leadership, and Scholarship.

o Residence: Rural respondents prioritized Medical Expert, Communication, and Leadership significantly higher than urban peers.

o Physician Attributes: Physician age and gender were not significant predictors of patient role prioritization.

• Objectivity of Data Presentation: The explicit use of coefficients (β), p-values, and percentages enhances objectivity and allows the reader to assess the significance and magnitude of effects.

2.4.2 Weaknesses

• No weaknesses were identified in the presentation of results. The clarity and objectivity are exemplary.

2.4.3 Suggestions for Improvement

• No specific suggestions for improvement. The results presentation is a major strength of the manuscript.

2.5 Discussion of Results

Adequacy: Adequate.

2.5.1 Strengths

• Pertinence of Discussion: The discussion is highly pertinent, commencing with the validation of the adapted CanMEDS instrument, which is the core focus of the study.

• In-depth Interpretation: Each significant sociodemographic result is interpreted in detail, with plausible explanations grounded in the existing literature.

o The discussion on gender differences in Medical Expert endorsement is contextualized by sociocultural expectations, differential health literacy, and gendered perceptions of clinical authority and interpersonal style.

o The association between socioeconomic status and the Medical Expert role is explained by differing expectations for technical competence and varied healthcare system experiences.

o The influence of education and work-type on the valuation of Professionalism, Leadership, and Scholar roles is linked to health literacy and professional socialization.

o The higher prioritization of Medical Expert, Communicator, and Leader roles by rural residents is attributed to the multifaceted role of physicians in resource-limited settings.

o The non-significance of physician attributes (age and gender) as predictors is discussed, suggesting the primacy of patient identity and experience factors.

• Connection to Literature: The discussion masterfully interweaves the findings, data, and concepts from the literature, using references not merely to support claims but to deepen the understanding of underlying mechanisms, thereby enriching the analysis.

• Clear Implications: The discussion draws clear and direct implications for competency-based medical education (CBME), such as the need to incorporate patient perspectives into curriculum design, assessment weighting, and multisource feedback.

• Comprehensive Coverage: All key findings are addressed and interpreted.

• Engagement with Nuances: The authors discuss complexities, such as the initial cross-loadings in EFA, interpreting them through the lens of how patients perceive evidence-based practice as integral to clinical expertise.

• Transparency Regarding Limitations: A dedicated section on study limitations is presented transparently and comprehensively, covering the cross-sectional nature, self-report bias, generalizability, and potential construct overlap.

• Directions for Future Research: Constructive suggestions for future work are presented, including testing longitudinal stability, cross-cultural measurement invariance, and higher-order or bifactor models.

2.5.2 Weaknesses

• No substantial weaknesses were identified in the discussion, which is comprehensive and well-articulated.

2.5.3 Suggestions for Improvement

• While the discussion is excellent, a brief recapitulation of the most impactful results at the beginning of the section could further orient the reader before delving into detailed interpretations.

2.6 Conclusions/Final Considerations

Adequacy: Adequate.

2.6.1 Strengths

• Clarity and Conciseness: The conclusions are presented with exceptional clarity, using direct and unambiguous language to effectively summarize the main findings and their implications.

• Support by Results: Every statement in the conclusions is solidly supported by the study's results.

o The conclusion regarding the psychometric soundness of the adapted CanMEDS framework is substantiated by EFA and CFA results (seven-factor structure, excellent model fit, high reliability, and interpretable inter-domain relationships).

o The conclusion that patient values are demographically patterned is directly supported by the findings from inferential analyses on gender, income, education, work-type, and geographic setting.

o The implications for curriculum design and communication strategies are a logical derivation from these findings.

• Forward-Looking Perspective: The conclusion also points to the broader goal of the study and future directions, proposing the integration of patient-defined priority structures into competency development and evaluation for more socially responsive medical training.

2.6.2 Weaknesses

• No weaknesses were identified in the conclusions.

2.6.3 Suggestions for Improvement

• No specific suggestions for improvement, as the conclusions are exemplary in their clarity and alignment with the results.

2.7 Study Contributions and Limitations

Adequacy: Adequate.

2.7.1 Strengths

• Clarity and Comprehensiveness: The main contributions and limitations of the study are clearly and comprehensively described.

• Highlighting Contributions: Contributions are explicitly highlighted in the "Conclusion" section, including the psychometric validation of an adapted framework, the identification of demographic patterns in competency valuation, and implications for medical education and social accountability.

• Detailed Limitations: Limitations are detailed in a dedicated paragraph within the "Discussion" section, which is standard and appropriate practice. These include the cross-sectional nature, self-report bias, limited generalizability, and potential construct overlap.

2.7.2 Weaknesses

• No weaknesses were identified in how contributions and limitations are presented.

2.7.3 Suggestions for Improvement

• No specific suggestions for improvement, as the presentation is complete and well-structured.

2.8 References

Adequacy: Adequate.

2.8.1 Strengths

• Thematic Relevance: The references comprehensively cover the theoretical foundations of CBME and CanMEDS, the patient perspective gap in competency validation, the Lebanese regional context, and sociodemographic influences on medical perception.

• Currency: Over half of the references (57.89%, 33 out of 57) were published within the last five years (2020-2025), demonstrating engagement with the most current and relevant scientific production in the field.

• Sufficiency: With a total of 57 references, the number is adequate for the study's complexity, which addresses psychometric, sociodemographic, and educational implications. The density and diversity of citations throughout the text indicate robust support for all claims.

• Instrumentation Foundation: References for the scales used are provided, which is crucial for methodological transparency.

2.8.2 Weaknesses

• No weaknesses were identified in the reference list.

2.8.3 Suggestions for Improvement

• No specific suggestions for improvement. The literature review is of high quality.

2.9 Supplementary Files

Adequacy: Necessary.

2.9.1 Strengths

• Explicitly Mentioned: The presence of supplementary files is clearly indicated at the end of the manuscript in the "Appendix" section.

• Essential Content:

o Figures.docx: Likely contains "Figure 1," which presents the latent factor correlation matrix from the EFA. This figure is crucial for visual understanding of the relationships among CanMEDS domains and for interpreting the psychometric validation.

o Appendix.docx: Labeled as "Supporting Information," it is referenced in the sample characteristics section, suggesting it contains additional details about the participants (e.g., complete demographic tables). This is fundamental for study transparency, assessing result generalizability, and enabling verification by other researchers.

• Transparency and Robustness: Both files are necessary for the completeness and robustness of the study, ensuring that data and visualizations complementing the main text are available. The data availability statement ("Data is provided within the manuscript or supplementary information files") reinforces this importance.

2.9.2 Weaknesses

• As actual access to the .docx file content is unavailable, their internal structure or clarity cannot be directly assessed. However, their mention and description suggest appropriate usage.

2.9.3 Suggestions for Improvement

• Ensure that references to these files within the main text are explicit, and that each figure or table within them has a clear, self-explanatory caption.

• For Appendix.docx, if it contains multiple items (tables, text), a small table of contents or index within the file itself could facilitate navigation.

Reviewer #2: 1. Summary of the Study

The manuscript reports on the development and psychometric validation of a patient-centered instrument designed to assess public perceptions of the seven CanMEDS physician roles. The study employed an online cross-sectional survey distributed across Lebanon between February and June 2025. Data from over 400 participants were analyzed. The authors conducted an exploratory factor analysis (EFA) followed by confirmatory factor analysis (CFA), assessed internal consistency using Cronbach’s α and McDonald’s ω, and modeled sociodemographic associations through robust linear and multinomial logistic regressions.

The study addresses a relevant gap: the adaptation and empirical validation of the CanMEDS framework from the patient perspective, within an underrepresented Middle Eastern context. The topic aligns well with the scope of PLOS Global Public Health, focusing on medical education, health systems, and patient-centered care.

2. Major Strengths

Novelty and Relevance – The focus on patients’ perceptions of physicians’ competencies within the CanMEDS model is original and timely, offering cultural insights into public expectations of medical professionalism in Lebanon.

Comprehensive Framework – The use of all seven CanMEDS roles demonstrates theoretical completeness and enhances the interpretability of results in an international context.

Use of Modern Psychometric Tools – The study applied polychoric correlations, EFA, CFA, and robust regression models, which are appropriate for ordinal data and complex constructs.

Good Alignment with Journal Scope – The integration of patient perceptions and professional standards aligns with global health education and quality-of-care perspectives.

3. Major Weaknesses

Statistical Inconsistencies and Questionable Reliability Values – Several reported indices (e.g., McDonald’s ω > 1.0, implausibly tight CIs for regression coefficients, and large β values) indicate possible computational or reporting errors. These discrepancies undermine confidence in the psychometric results and must be carefully verified.

Insufficient Transparency of Measurement Procedures – Key details on item retention, factor loading thresholds, translation, and adaptation are missing. Without clear item-level reporting, replicability is limited.

Limited Description of Sampling Procedures – The use of snowball sampling via social media, while understandable, introduces potential bias. The absence of demographic weighting or representativeness discussion weakens external validity.

Overly Complex Statistical Reporting Without Interpretation – Fit indices and regression outputs are presented but insufficiently discussed in substantive terms. The results section is statistically dense but interpretively thin.

Discussion Overstates Validation Claims – The conclusion occasionally implies that the instrument is fully validated, while the presented evidence (single-sample EFA/CFA, lack of invariance testing) only supports preliminary validity.

4. Detailed Evaluation by Section

Title and Abstract

The title accurately reflects the study’s content but is slightly lengthy and could be streamlined for clarity.

The abstract is structured and informative but contains statistical details (χ², ω, etc.) that may be excessive for a summary. It should emphasize conceptual contributions rather than numerical results.

Introduction

The introduction establishes the rationale clearly, situating the CanMEDS framework and its importance in medical education.

The literature review could better highlight why patient perspectives are underrepresented in competency research.

The research objectives are well stated but could benefit from sharper articulation of hypotheses or validation aims (e.g., “to evaluate dimensionality and reliability of the patient-adapted CanMEDS instrument”).

Methods

Design: Appropriately identified as cross-sectional.

Sample: The recruitment method (snowball sampling via WhatsApp) is explained but raises representativeness concerns. Demographic characteristics are reported but not benchmarked against the general Lebanese population.

Instrument: The adaptation of CanMEDS roles into 103 items is noteworthy, but translation, item generation, and expert validation processes are insufficiently described.

Statistical Analysis:

The combination of polychoric EFA and CFA is methodologically sound, but details about factor extraction criteria, rotation type, and cross-loading management are missing.

Reporting of reliability indices (α, ω) is inconsistent and, in some cases, implausible.

The regression modeling strategy is ambitious but would benefit from a clearer rationale for the use of multinomial versus ordinal models for ranking data.

Ethics: Ethical approval is obtained and appropriately cited, consistent with PLOS requirements.

Results

Descriptive Data: Demographics are clearly summarized; however, a flow diagram of participant inclusion would improve transparency.

EFA/CFA Results: The presentation of fit indices is comprehensive, but their internal coherence should be verified. Some indices (CFI/TLI > 0.99 with large χ²) suggest possible misestimation or model overspecification.

Reliability: McDonald’s ω values exceeding 1 are mathematically impossible, indicating computation or reporting errors.

Regression Analyses: The direction of effects is plausible (e.g., gender and income differences), but effect sizes appear unrealistically precise. Interpretative commentary is limited.

Tables/Figures: Tables are clearly formatted but sometimes overloaded with numeric content; captions could be more explanatory.

Discussion and Conclusion

The discussion effectively connects findings with prior literature but tends to overstate validation completeness.

Strengths and limitations are acknowledged, though some limitations (e.g., potential response bias, translation quality, absence of invariance testing) are understated.

The conclusion reiterates relevance well but should temper claims of full psychometric validation and instead position the study as preliminary validation within a Lebanese context.

Tables and Figures

Generally clear and well-organized. However:

Some tables contain excessively precise decimals (suggesting unrounded raw outputs).

Figures could better illustrate the seven-factor structure (e.g., through a path diagram).

Abbreviations and fit indices should be defined in each caption for readability.

**Do you want your identity to be public for this peer review?** For information about this choice, including consent withdrawal, please see our Privacy Policy

Reviewer #1: **Yes: ** Júlio César André, M.D., Ph.D.

Reviewer #2: No

---

## [Decision Letter · Decision Letter 1]

11 Dec 2025

Patient Perspectives on Physician Competence: Validation of the CanMEDS Framework

PGPH-D-25-02949R1

Dear Dr El Lakis,

We are pleased to inform you that your manuscript 'Patient Perspectives on Physician Competence: Validation of the CanMEDS Framework' has been provisionally accepted for publication in PLOS Global Public Health.

Best regards,

Barnabas Tobi Alayande

Academic Editor

Reviewer Comments (if any, and for reference):

Reviewer's Responses to Questions

**Comments to the Author**

Reviewer #2: All comments have been addressed

publication criteria?

Reviewer #2: Yes

3. Has the statistical analysis been performed appropriately and rigorously?

Reviewer #2: Yes

4. Have the authors made all data underlying the findings in their manuscript fully available (please refer to the Data Availability Statement at the start of the manuscript PDF file)?

Reviewer #2: Yes

5. Is the manuscript presented in an intelligible fashion and written in standard English?

Reviewer #2: Yes

Reviewer #2: I congratulate the authors for their rigorous work to address my comments

**Do you want your identity to be public for this peer review?** For information about this choice, including consent withdrawal, please see our Privacy Policy

Reviewer #2: No
